# Gluconate Kinase Is Required for Gluconate Assimilation and Sporulation in *Cryptococcus neoformans*

Andrew J. Jezewski,[a] Sarah R. Beattie,[a] Katy M. Alden,[a] (ID) Damian J. Krysan[a,b]

[a]Department of Pediatrics, Carver College of Medicine, University of Iowa, Iowa City, Iowa, USA
[b]Microbiology/Immunology, Carver College of Medicine, University of Iowa, Iowa City, Iowa, USA

**ABSTRACT** *Cryptococcus neoformans* is an environmental yeast and an opportunistic human pathogen. The ability to cause disease depends on the ability to adapt to the human host. Previous studies implicated infectivity-related kinase 3 (*IRK3*, CNAG_03048) as required for establishing an infection. We genetically and biochemically characterized *IRK3* as a gluconate kinase and propose the name *GNK1*. This metabolic enzyme utilizes gluconate to produce 6-phosphogluconate as part of the alternative oxidative phase of the pentose phosphate pathway (AOXPPP). The presence of *GNK1* confirms that the AOXPPP is present and able to compensate for loss of the traditional OXPPP, providing an explanation for its nonessentiality. *C. neoformans* can utilize gluconate as an alternative carbon source in a *GNK1*-dependent manner. In our efforts to understand the role of *GNK1* in host adaptation and virulence, we found that *GNK1*-deficient mutants have variable virulence and carbon dioxide tolerance across multiple strains, suggesting that second site mutations frequently interact with *GNK1* deletion mutations. In our effort to isolate these genetic loci by backcrossing experiments, we discovered that *GNK1*-deficient strains are unable to sporulate. These data suggest that gluconate metabolism is critical for sporulation of *C. neoformans*.

**IMPORTANCE** *Cryptococcus neoformans* is a fungal pathogen that contributes to nearly 180,000 deaths annually. We characterized a gene named *GNK1* that appears to interact with other genetic loci involved with the ability of *C. neoformans* to act as a pathogen. While these interacting genetic loci remain elusive, we discovered that *GNK1* plays roles in both metabolism and mating/sporulation. Further interrogation of the mechanistic role for *GNK1* in sexual reproduction may uncover a larger network of genes that are important for host adaptation and virulence.

**KEYWORDS** *Cryptococcus neoformans*, metabolism, carbon dioxide, gluconate kinase

Cryptococcus neoformans is an environmental yeast that causes severe disease in immunocompromised and immunosuppressed individuals (1). Healthy humans have multiple defense mechanisms for controlling a *Cryptococcus* infection, including but not limited to nutrient deprivation, oxidative insults, carbon dioxide stress, and heat stress (2). Metabolic processes allow for rapid response to cellular and environmental signals to provide the biomass, energy requirements, or secondary signals that are required to overcome a stress (3–6). As is characteristic of many fungi, *C. neoformans* is armed with metabolic plasticity that allows for a robust response to diverse growth conditions (3–6). A recently identified virulence-related adaption of *C. neoformans* is its ability to overcome carbon dioxide stress (7). To understand the mechanisms that play a role in carbon dioxide tolerance, we identified the gene named infectivity-related kinase 3 (*IRK3*, CNAG_03048) in a screen for genes involved in carbon dioxide tolerance using a systematic library of kinase deletion mutants (8). Previously, deletion of this gene resulted in no clear phenotype *in vitro*, but it led to a failure to establish infection in the lungs of mice (8). We set out to characterize the function and role of this gene in the context of infection and carbon dioxide

Address correspondence to Damian J. Krysan, damian-krysan@uiowa.edu.

The authors declare no conflict of interest.

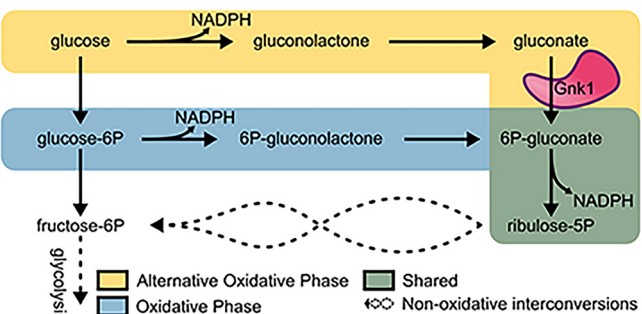

**FIG 1** *GNK1* functions in the alternative oxidative pentose phosphate pathway. Schematic highlights the differences between the tradition oxidative phase of the pentose phosphate pathway (blue) and the alternative oxidative phase of the pentose phosphate pathway (yellow) with their shared final step (green). Steps that produce reducing equivalents are indicated by their production of NADPH. The metabolic role for Gnk1 is indicated by the labeled illustration (pink).

tolerance. Although it does not appear to have a direct role in carbon dioxide tolerance, we discovered that it functions as a metabolic kinase involved in the alternative oxidative phase of the pentose phosphate pathway (AOXPPP, Fig. 1). Thus, we have renamed this gene and now refer to it as a 6-phosphogluconate kinase (*GNK1*, CNAG_03048).

The primary role of the pentose phosphate pathway (PPP) is anabolic, where five carbon sugars are generated for nucleotide and histidine biosynthesis (note: histidine biosynthesis is not present in mammals) (9). The PPP also has a catabolic role and is sometimes referred to as the hexose monophosphate shunt (HMP) (9). The PPP responds to the oxidative insults encountered by *C. neoformans* during an infection and provides building blocks required for growth/replication (10). In its role as the HMP shunt, glucose is oxidized in the first half of the pathway to generate reducing equivalents in the form of NADPH. Carbon then re-enters glycolysis to further support the energy needs of the cell through a series of nonoxidative interconversions. The oxidative phase of the PPP is carried out by glucose-6-phosphate dehydrogenase (*ZFW1*, CNAG_03245), 6-phosphogluconolactonase (CNAG_04753), and 6-phosphogluconate dehydrogenase (CNAG_04099, CNAG_07561), where NADPH is produced at each of the dehydrogenase steps (blue highlight, Fig. 1). An ancient metabolic alternative to glycolysis involves the Entner-Doudoroff (ED) pathway, which maximizes reducing power at the expense of ATP production albeit at a reduced overall protein requirement (11).

The first part of this alternative consumption route of glucose includes an AOXPPP. Many organisms, including fungi, have maintained the AOXPPP despite lacking the machinery or reliance on the remaining steps of the ED pathway (12). The AOXPPP is carried out by glucose oxidase (CNAG_06081) and gluconolactonase (Lhc1p, CNAG_04753) acting parallel to the traditional oxidative phase of the PPP (yellow highlight, Fig. 1). Carbon then re-enters the PPP as 6-phosphogluconate through Gnk1 activity, and the final step of the alternative oxidative phase is completed by 6-phosphogluconate dehydrogenase, which is part of the traditional PPP. Gnk1 acts as a metabolic kinase that utilizes ATP to phosphorylate gluconate and generate 6-phosphogluconate. As mentioned above, the AOXPPP is redundant to the traditional oxidative phase of the PPP in *C. neoformans*. It remains unclear why the redundancy of the AOXPPP to the traditional oxidative phase of the PPP would be so highly conserved across fungi and higher eukaryotes, including mammals.

Previous studies investigating the role of the PPP in *C. neoformans* found that the traditional PPP entry was neither essential nor required for generating reducing equivalents (10). This observation is in contrast to the fact that many other organisms utilize the PPP as the major producer of reducing equivalents. As a possible explanation for this observation, Brown et al. postulated that the AOXPPP may be present in *C. neoformans* and may be sufficient to generate reducing equivalents (10). The presence of the AOXPPP is necessary for other organisms to utilize gluconate, glucuronate, and/or galacturonate as alternative carbon sources (13). These sugar-derived carboxylic acids can be present in host niches for organisms that maintain the AOXPPP and, hence, may be relevant to virulence. For

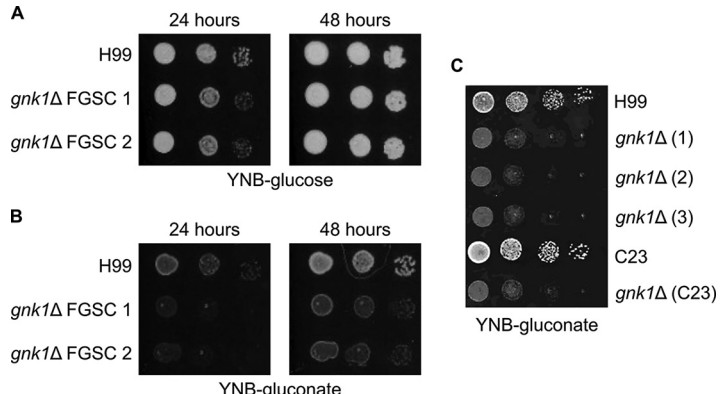

**FIG 2** *C. neoformans* utilizes gluconate as a carbon source in a *GNK1*-dependent manner. Spot dilutions of gluconate kinase deficient mutants grown on YNB media with either (A) 2% glucose or (B) 2% gluconate supplied as the carbon source. (C) Multiple independent isolates of *GNK1* deletion mutants in the H99 background as well as a *GNK1* deletion from a separate clinical isolate (C23) were incubated on YNB-gluconate. The plating involves a 1:10 dilution series from starting an overnight culture adjusted to an $OD_{600}$ = 1. FGSC indicates strains obtained from the Bahn Laboratory's deletion collection available from the <u>F</u>ungal <u>G</u>enetics <u>S</u>tock <u>C</u>enter (8).

example, macrophages restrict nutrients, including carbon, to pathogens taken up within the phagolysosome (2). Like *C. neoformans*, *Salmonella typhimurium* must survive the *in vivo* niche of the phagolysosome within macrophages (14). *Salmonella* has been shown to primarily utilize gluconate as a carbon source in macrophages (14).

Prior to the work described here, the role of gluconate kinase in *C. neoformans* biology had not been described. While *GNK1* deletion was screened for its role in infectivity and virulence as part of the kinase deletion library by Lee et al., its specific role remained unclear (8). This competitive fitness study used pooled libraries of barcoded strains that implicated *GNK1* deletion mutants as having a defect in infectivity but not virulence (15). Our studies highlight a far more complex role for *GNK1* in driving defects in infectivity and/or virulence. While its role in virulence remains unclear due to the variability of independent isolates of *GNK1* deletion mutants in mouse models of infection, we have confirmed the metabolic role for this gene product as a gluconate kinase and its function is required for sporulation.

## RESULTS

**Alternative oxidative pentose phosphate pathway is conserved in pathogenic fungi.** While the ED pathway has long been lost as a primary route of glycolysis, many organisms retain parts of this pathway. At a minimum, we identified a glucose oxidase, lactonohydrolase, and gluconate kinase in the pathogenic fungi *Cryptococcus*, *Aspergillus*, *Histoplasma*, *Coccidioides*, *Candida*, and *Mucor*. These enzymes comprise the initial steps of the ancient ED pathway and appear to be co-opted as an alternative entry into the PPP (Fig. 1). Alignment of the gluconate kinases from each of these genera along with *E. coli* and humans indicates the sequences of these proteins are conserved across the species (>30% similarity; Table S1 in the supplemental material). While there was a mild increase in similarity between fungal species relative to other organisms, these similarities are not phylogenetically discriminated from the human and *E. coli* gluconate kinases (Fig. S1).

**Gluconate is a viable carbon source dependent on Gnk1 activity in *C. neoformans*.** Prokaryotes that maintain the ED pathway are thought to do so because their environmental niches contain high levels of acidic sugars such as gluconate, glucuronate, and galacturonate (13). Because Gnk1 is homologous to other gluconate kinases, we hypothesized that it may play a role in utilizing gluconate as a carbon source in *C. neoformans*. To test this hypothesis, we incubated *gnk1Δ* strains on minimal media containing either glucose or gluconate as the primary carbon source. The growth of *gnk1Δ* strains was only minimally affected when glucose was the sole carbon source (Fig. 2A). Importantly, the reference strain H99 as well as a clinical isolate of *C. neoformans* utilize gluconate as a sole

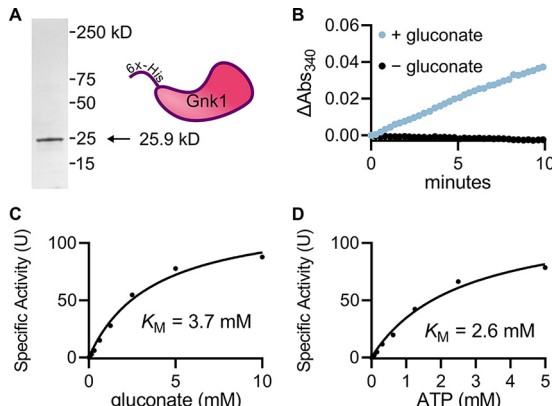

**FIG 3** Biochemical characterization of *C. neoformans* Gnk1. (A) Coomassie-stained SDS-PAGE gel of heterologously expressed *Cn*Gnk1. (B) Kinetic assay showing gluconate-dependent enzyme activity. Michaelis-Menten kinetics of Gnk1p substrates gluconate (C) and ATP (D) with the other substrate at saturating concentrations. The Michaelis-Menten constants are apparent because of the bi-substrate nature of the reaction. Curves are representative of experimental duplicates each containing technical duplicates and plotted using GraphPad Prism. Confidence intervals of Michaelis Menten constants reported in text.

carbon source. The ability to utilize gluconate as a carbon source was *GNK1*-dependent in multiple independently derived deletion mutants and in a separate clinical isolate (Fig. 2C). Thus, *C. neoformans* can utilize gluconate as a carbon source and Gnk1 is required for its assimilation.

**Recombinant *C. neoformans* Gnk1 has gluconate kinase activity.** To confirm that *GNK1* encodes a gluconate kinase, we biochemically characterized recombinant *Cn*Gnk1. Heterologous expression of an N-terminal 6×-His-tagged *Cn*Gnk1 in *E. coli* followed by immobilized metal affinity chromatography yielded pure (>95%) recombinant protein (Fig. 3A). To characterize Gnk1, we used a spectrophotometric, continuous coupled assay where the enzyme substrates gluconate and ATP react in the presence of an excess of the recombinant accessory enzyme 6-phosphogluconate dehydrogenase (6PGDH) and NADP. Production of NADPH serves as an absorbance readout of Gnk1 activity (Fig. S2A). The activity is both substrate and enzyme dependent (Fig. S2B), validating the assay. We also defined the linear range of the assay with respect to protein content and reaction time (Fig. 3B, Fig. S2C). Consistent with its protein sequence and the phenotypic behavior of its deletion mutant, recombinant *Cn*Gnk1 exhibits gluconate kinase activity (Fig. 3B). We also measured the apparent Michaelis-Menten constants for each substrate (ATP and gluconate); these are apparent constants because the reaction involves two substrates, but we will refer to the values as $K_M$ for clarity. The apparent Michaelis-Menten constants for the two substrates are ATP $K_M$ = 2.6 mM, 95% CI (1.9, 3.7), and gluconate $K_M$ = 3.7 mM, 95% CI (2.7, 5.2) (Fig. 3C and D) and indicate that Gnk1, like many enzymes involved in carbon metabolism, has relatively low affinity for its substrates.

To assess whether Gnk1 might be able to accommodate alternative sugars, we competed glucose, mannitol, and galactose against gluconate using the same assay reported above. Competing substrates were supplied in excess (10 mM) compared to gluconate (0.5 mM). Only mild reduction in activity was observed for glucose (~ 40%) and less so for mannitol (~20%) and galactose (~20%) (Fig. S3). This supports a strong preference for gluconate as the primary sugar substrate.

**Gluconate kinase deletion induces context dependent carbon dioxide intolerance.** We originally identified *GNK1* as a gene of interest due to its apparent role in determining carbon dioxide tolerance. As previously described, carbon dioxide is a host stress that must be overcome to exhibit virulence (7). Two-independent *GNK1* deletion mutants show reduced growth at host-relevant concentrations of carbon dioxide (5%) compared to ambient air conditions (Fig. 4A). Because these mutants (FGSC 1 and FGSC 2) were generated as part of a large-scale library construction project, we sought to confirm the carbon dioxide tolerance phenotype by generating independent *gnk1*Δ

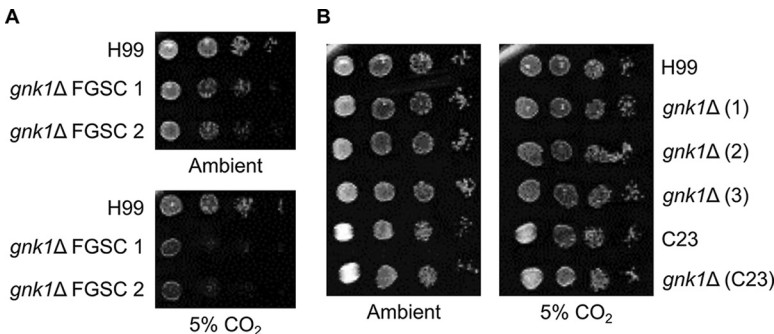

**FIG 4** *GNK1* deletion mutants do not have consistent effects on carbon dioxide tolerance. Spot dilutions of *GNK1* deletion mutants from the fungal genetics stock center (FGSC) incubated on YNB with 2% glucose media with either ambient or 5% carbon dioxide (A). Multiple, independently generated isolates derived from H99 and a *GNK1* deletion from a separate clinically derived strain (C23) grown on YNB with either ambient or 5% carbon dioxide (B). The plating involves a 1:10 dilution series from starting an overnight culture adjusted to an $OD_{600} = 1$.

mutants in the same H99 background and in a separate clinical isolate (C23) (16). Surprisingly, none of a set of independent *gnk1*Δ mutants showed altered growth in the presence of carbon dioxide. (Fig. 4B). All strains were independently PCR confirmed for deletion of *GNK1* (data not shown). In contrast, all *gnk1*Δ strains exhibited decreased growth on gluconate medium as shown in Fig. 2, indicating that these strains were lacking Gnk1 activity. Together, these observations suggest that the carbon dioxide phenotype can be decoupled from gluconate kinase deletion and thus may be the result of a second site mutation in the initial library derived strains.

**Virulence defects in *GNK1* deletion mutants are independent of reduced carbon dioxide tolerance.** As part of their comprehensive phenotyping of *C. neoformans* kinases, Lee et al. found that the *gnk1*Δ mutants were less fit during pulmonary infection based on a signature tag competition experiment (8). This same group found, in a second competition experiment using the tail vein route of infection, that *GNK1* deletion mutants were not deficient in dissemination (15). We directly tested the virulence of the library- and independently-derived *GNK1* deletion mutants in both the pulmonary and dissemination mouse models using the same A/J mouse strain background. Consistent with the infectivity defects reported by Lee et al., we found that both isolates of the *GNK1* deletion mutants from the kinase deletion set exhibit a virulence defect in the pulmonary model, log-rank (Mantel-Cox) test $P < 0.01$ (Fig. 5A). In contrast to their lack of an infectivity phenotype in the dissemination model, we observed a virulence defect for these in the dissemination model (Fig. 5B), log-rank (Mantel-Cox) test $P < 0.0001$. Interestingly, surviving mice infected with the *gnk1*Δ mutant showed significant fungal burden in the brain, indicating that the mutants had disseminated to the brain but did not cause terminal disease (Fig. 5C).

Next, we tested the virulence of our carbon dioxide-tolerant *gnk1*Δ deletion mutants (Fig. 5D). While decreased growth on gluconate was conserved across deletion mutants, carbon dioxide tolerance was not. We had hoped that testing virulence across these phenotypically divergent strains would help us determine if the virulence defect observed was linked to either altered gluconate utilization or carbon dioxide tolerance. To our dismay, we observed a mixed virulence phenotype for these strains such that virulence was neither consistent with decreased gluconate utilization nor altered carbon dioxide tolerance (Fig. 5D). The Bahn kinase deletion set derived *GNK1* deletion mutant FGSC 1, and only one of our deletion mutants *gnk1*Δ 5 exhibited a virulence defect, log-rank (Mantel-Cox) test $P < 0.001$. To determine if these discrepancies were due to experimental error, an inexplicable genetic reversion, or strain contamination, we isolated culture directly from endpoint mice and performed Southern blots and confirmed their *GNK1* deletion genotype (Fig. 5E).

To further complicate matters, the *gnk1*Δ mutant derived from the C23 clinical isolate also showed an intermediate virulence phenotype, where only a subset of mice experienced delayed onset of illness and one mouse never displayed symptoms (Fig. 5F). Thus, we

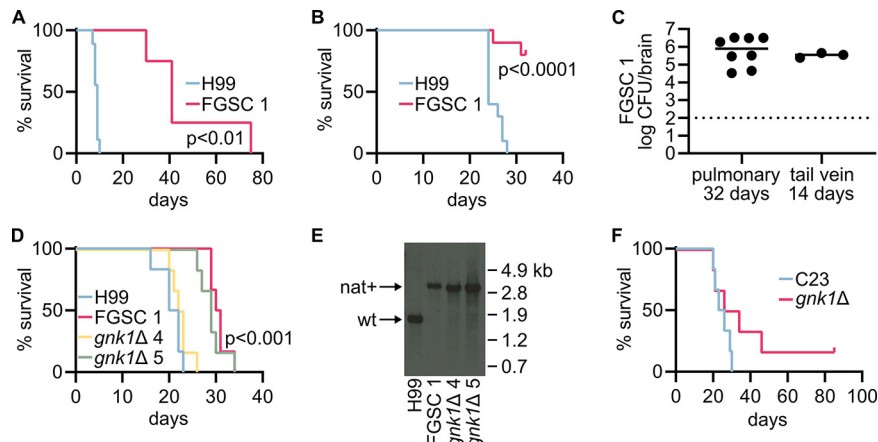

**FIG 5** *GNK1* deletion mutants do not have consistent effects on virulence. Survival of A/J mice infected via the tail vein with the H99 reference strain (*n* = 10) or *gnk1*Δ FGSC 1 from the kinase deletion library (*n* = 4) (A). Survival of A/J mice infected via the pulmonary route with H99 (*n* = 10) or *gnk1*Δ FGSC 1 (*n* = 10) (B). Brain burden of *gnk1*Δ FGSC 1 in A/J mice on day 32 of a pulmonary inoculation or day 14 of a tail vein inoculation (C). Survival of A/J mice infected via the pulmonary route with indicated strains, *n* = 6 for all strains (D). Southern blot of DNA isolated from *C. neoformans* recovered from mice in panel E. (F) Survival of A/J mice infected via the pulmonary route with a clinical isolate C23 (*n* = 6) and the C23-derived *gnk1*Δ mutant (*n* = 6).

cannot attribute deletion of *GNK1* to a consistent, direct effect on virulence. The most likely explanation is that the virulence phenotype is dependent on both loss of Gnk1 function and a second mutation in the strain. Unfortunately, complementation will not resolve this issue, because re-introduction of *GNK1* would be expected to restore function regardless of the presence or absence of the mutation.

**GNK1 deletion is required for sporulation.** As discussed above, the phenotypic heterogeneity resulting from *GNK1* deletion suggests that a second site mutation may be interacting with *gnk1*Δ to drive the phenotypes. To identify this potential second site mutation, we crossed the *gnk1*Δ isolates with their respective parent strain. To our surprise, none of the crosses with *GNK1* deletion mutants produced spores, while the parental strains sporulated readily as expected (Fig. 6A and B). The *gnk1*Δ deletion mutants formed hyphae and basidia (Fig. 6B insets), indicating that mating and meiosis was likely occurring. However, despite extending the incubation time up to 8 weeks post-plating, no spores could be identified in crosses from any of the independent *gnk1*Δ mutants. The lack of sporulation prevents genetic isolation of potential interacting mutations. Regardless, we are able to unify a phenotype beyond the gluconate utilization across the *GNK1* deletion mutants. One possibility for the lack of sporulation and the genetic variation associated with loss of Gnk1 function is that the reduced activity of the PPP pathway leads to changes in nucleotide homeostasis. If that were the case, then *gnk1*Δ mutants might be expected to show increased genetic instability in general. To test this possibility, we examined their sensitivity to UV exposure and methyl methanesulfonate (MMS) treatment. However, the *gnk1*Δ mutants grew similar to wild type in the presence of these genomic stressors (data not shown). At this point, further work will be needed to understand the mechanistic basis for the role of Gnk1 during sporulation. The virulence contributions of Gnk1 remain enigmatic and might be best addressed with a genetic system in which regulated depletion of the gene could be achieved during infection; however, at present those methods are largely unavailable for *C. neoformans*.

## DISCUSSION

*C. neoformans* adapts to a broad range of environments that vary in nutrient availability, growth restriction, and predation (2). Some isolates/strains of *C. neoformans* are better adapters or have an enhanced innate ability to survive in the human host (16). This is supported by the presence of pathogenic and nonpathogenic strains of

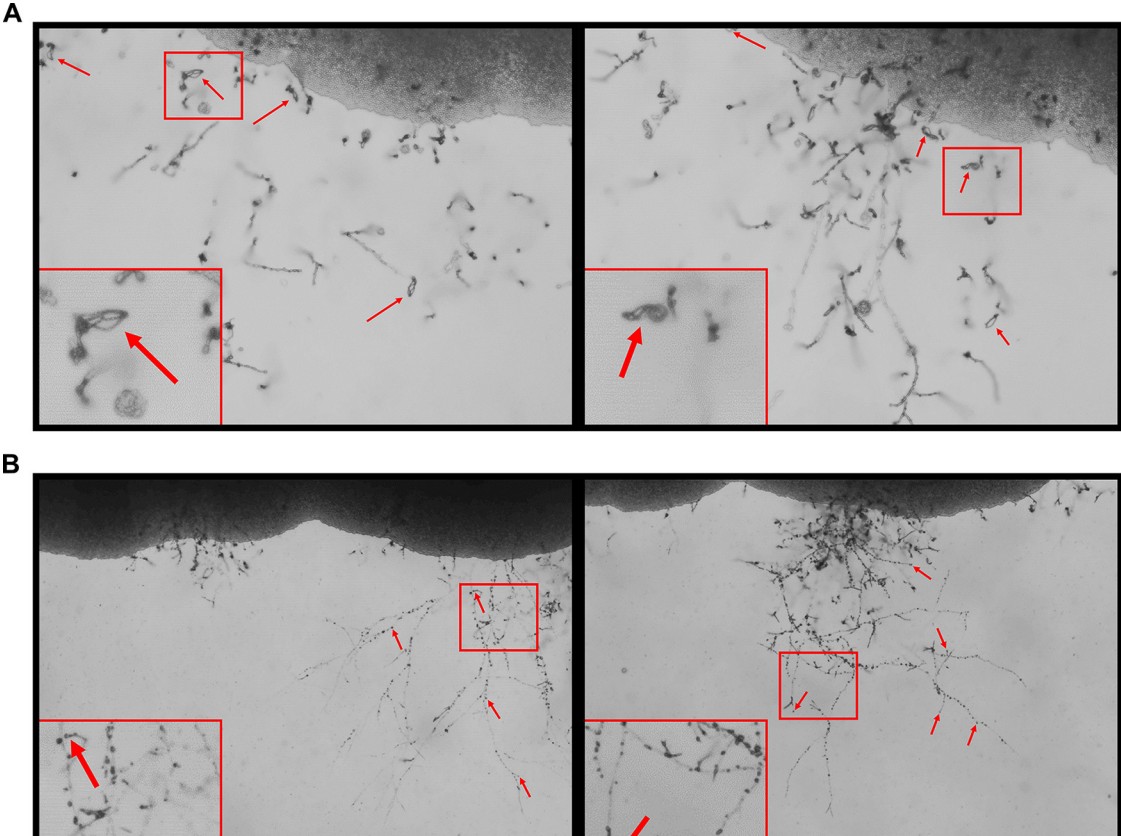

**FIG 6** *GNK1* deletion mutants have a sporulation defect. Representative images from H99a crossed with KN99α (A) or a *gnk1*Δ mutant crossed with H99a (B) 8 weeks postplating on V8 agar. Large red boxes are enlargements of the regions outlined in red. Red arrows highlight spore chains present in the H99a × KN99α crossing. The black arrows in B indicate the presence of basidia without spore chains.

*C. neoformans* (16). Characterizing the mechanisms by which *C. neoformans* adapts to the host is essential for understanding how this opportunistic pathogen overcomes host defenses and drives disease. We set out to characterize the role of *IRK3* in host adaptability and virulence. Our studies show that, in fact, *IRK3* codes for a gluconate kinase based on our biochemical validation and genetic studies that show a highly reproducible dependence on *GNK1* for gluconate utilization. Thus, we propose that *IRK3* be renamed *GNK1*.

While gluconate is considered an "alternate" carbon source, it should be put into context that gluconate is widely available in nature. As an oxidized form of glucose, many environmental niches are likely to contain mM levels of gluconate as a primary carbon source and in many cases only second to glucose. In fact, gluconate has been reported to be as high as one-third of glucose serum concentrations, with the average human producing as much as 25–30 g per day (17). As an environmental yeast and accidental pathogen, perhaps *C. neoformans* has an environmental requirement for maintaining gluconate utilization machinery, and this provides some advantage in its environmental niche. These readily bioavailable gluconate concentrations provide context to the relatively low affinity of Gnk1 (3.7 mM $K_M$) for gluconate. Given that gluconate kinases participate in the AOXPPP as an "alternate" pathway, it seems plausible that Gnk1 would function only when its substrates were abundant. However, it is possible this enzyme is regulated posttranslationally to change its affinity for substrates under changing AOXPPP needs or when gluconate is a predominant or sole-carbon source.

Beyond characterizing the biochemical role of *GNK1*, we quickly encountered a problem in trying to answer questions about its role in virulence and host adaptation.

Despite multiple independently derived *GNK1* knockout strains, almost no other phenotype segregated with the *GNK1* genotype. We considered multiple possible explanations for this observation. First, strain background differences interact with *GNK1*. While the genetic background segregated nicely for carbon dioxide tolerance, this did not explain differences in virulence. Therefore, it is not likely that a second site background mutation either masked the phenotype in our independently derived strains or unveiled the phenotype in the FGSC library derived strains.

Second, *GNK1* deletion results in a genetic instability that gives rise to independently acting genetic loci driving the phenotypes observed. If a genetic instability were driving these differences, we would have expected to observe an increased sensitivity for DNA damaging agents, but we did not. Third, *GNK1* deletion induces a fitness cost that can be resolved in more than one way. This explanation seems to fit well with our observations and indicates the potential for not one but two, three, or even more possible interactors that drive phenotypes apparently linked to *GNK1*. It also complicates the approaches we would consider for identifying these additional genetic loci and whether *GNK1* is required for the observed phenotype.

Given that our observations support multiple genetic interactors, the least helpful approach would be to perform complementation of *GNK1*. We might learn that *GNK1* is dominant to whatever genetic interactors are specifically present in the strains that display a phenotype. However, those conclusions would only apply to strains that showed complementation and would still leave the identity of the genetic interactors a mystery. An alternative approach may have been to perform whole genome sequencing; however, given that multiple genetic loci are likely at play, we would not be looking for a single common locus across the *GNK1* deletion strains. This could result in multiple candidates with no clear answer. A more appropriate approach would be to perform sexual backcrosses where progeny could be segregated by phenotype. This would simultaneously allow us to address two major questions regarding these genetic interactions: (i) is *GNK1* linked or able to be decoupled from the other genetic loci, and (ii) what are the genetic loci? The ability to generate dozens of progeny that are first segregated by phenotype then followed with whole genome sequencing would greatly simplify identifying the genetic interactors. Unfortunately, we found a completely unexpected phenotype that also perfectly segregated with the *GNK1* genotype. All *GNK1* deletions were unable to sporulate when crossed with a WT strain. This inability to sexually reproduce eliminated the pertinent approach to identifying the genetic loci driving our observed phenotypes. It did, however, introduce an interesting secondary role for *GNK1*.

It appears *GNK1* is required for sporulation. Clearly, extensive additional studies will be needed to identify the mechanism of the sporulation defect observed for the *gnk1Δ* mutants. The presence of hyphae and basidia suggests that the early stages of sexual reproduction are proceeding while the late stages are not. Perhaps products derived from gluconate uptake act as a signal for sporulation. We are less inclined to think gluconate accumulation acts as a sporulation suppression signal given that the sporulation medium is primarily V8 juice, which contains vitamins and minerals supplied with gluconate as the counter-ion. There have been several regulators identified for *C. neoformans* sexual reproduction, but fewer that are specifically required for sporulation. Previous works have identified two transcription factors, Crz1 and Zfp1, and two RNA binding proteins, Csa1 and Csa2, that were required for sporulation (18, 19). Additionally, a secreted protein Fad1 is required for proper spore maturation (19). Another metabolic enzyme that is required for sporulation is the carbonic anhydrase Can2. This enzyme reversibly converts carbon dioxide to bicarbonate. This seemingly draws a connection between *GNK1* and the carbon dioxide sensitivity phenotype, but the sporulation defect of Can2 deletion mutants is mainly present under high carbon dioxide, and these mutants are able to form spores in ambient air just as our mating assays were performed (20).

We encountered major difficulties in identifying what appear to be readily produced genetic interactors of *GNK1*. These interactions are important for virulence and host adaptation. While we were unable to identify these interacting genetic loci, we

can report the first biochemical and genetic confirmation of the AOXPPP. We can also report that gluconate is a viable carbon source for *C. neoformans* and that uptake of gluconate is dependent on the presence of the gluconate kinase Gnk1. Finally, *GNK1* is required for sporulation, although future studies will be required to determine the mechanism driving this phenotype.

## MATERIALS AND METHODS

**Strains, media, and growth conditions.** All strains were maintained in glycerol stocks kept at −80°C and recovered on YPD plates. All experiments were performed, unless otherwise stated, using cultures grown overnight shaking at 220 rpm at 30°C in 2 mL liquid YPD. *GNK1* (CNAG_03048) deletion strains were obtained from the FGSC as part of the kinase deletion set constructed by the Bahn laboratory (8); the gene was previously named *IRK3* (8, 21). Spotting assays were performed using 3 $\mu$L of 1:10 serial dilutions of a culture adjusted to an $OD_{600}$ = 1. Alternative carbon source assimilation was performed in minimal media (YNB) plates supplemented with either 2% gluconate or 2% glucose as indicated. Carbon dioxide tolerance was tested in a standard tissue culture incubator in a 5% $CO_2$ atmosphere.

**Strain construction.** *GNK1* knockout strains were generated by replacing the gene locus (CNAG_03048) with the nourseothricin acetyltransferase-resistance cassette (NAT) using the CRISPR TRACE method (22). As described by this method, two guides were generated to cut the 5′ and 3′ ends of the gene using primers 5′-AACGGATGAGTGCAAGCCAAGTTTTAGAGCTAGAAATAGCAAG-3′ and 5′-TTGGCTTGCACTCATCCGTTACAGTATACCCTGCCGGTG-3′ for the 5′ guide; and 5′-GTGACTGTGGCGAACATAGAGTTTTAGAGCTAGAAATAGCAAG-3′ and 5′-TCTATGTTCGCCACAGTCACACAGTATACCCTGCCGGTG-3′ for the 3′ guide. The repair construct was amplified using primers 5′-GTAAAACGACGGCCAGTGAGC-3′ and 5′-CAGGAAACAGCTATGACCATG-3′ from gDNA isolated from *GNK1* deletion strains in the kinase deletion library described above. All CRISPR components were PCR purified and transformed into either H99 or the clinical isolate C23 via electroporation using the "Pic" setting on a Bio-Rad Micropulser (16). Transformants were selected on YPD-NAT agar plates and genotyped for correct integration via PCR using diagnostic primers *IRK3* SO and B79 as previously described (8). Mutants utilized in subsequent animal experiments were further confirmed via Southern blot where isolated gDNA from endpoint recovered yeast cells was digested using ClaI and BsgI restriction enzymes and probed using a DIG labeled PCR product generated using primers 5′-TCACTGGCCGTCGTTTTACCTAAAAACGGAGCGGAAG-3′ and 5′-ATGGTCATAGCTGTTTCCTGCGAACTTCTCAAGCAACG-3′.

**Enzyme expression and purification.** The enzyme expression construct was provided by the Seattle Structural Genomics Center for Infectious Disease (SSGCID) reference ID CrneC.19317.a.B2.GE40908. The expression plasmid was transformed into BL21(DE3) Rosetta *E. coli*. Ampicillin resistant transformants were used to inoculate a 5 mL overnight culture of LB where 3 mL was used to inoculate a 1L culture of LB with continued ampicillin selection pressure. Once the shaking flask culture reached an $OD_{600}$ = 0.6, 1 mM isopropyl-$\beta$-D thiogalactopyranoside (IPTG) was added to induce expression at 37°C with continued shaking for 2 h. Pelleted cells were then resuspended in lysis buffer (1 mM PMSF, 1 mM DTT, 0.25 $\mu$L/L benzonase, 1 mg/mL lysozyme, 10 mM Tris-HCl pH 7.5, 20 mM imidazole, 1 mM $MgCl_2$, 200 mM NaCl) on ice and sonicated for two cycles of 3 min on, 3 min off. Cleared lysates were then mixed with washed Ni-agarose beads and allowed to flow through a column followed by 1 column volume (CV) of wash buffer (20 mM imidazole, 20 mM Tris-HCl pH 7.5, 150 mM NaCl), and 10 mL of elution buffer (300 mM imidazole, 20 mM tris-HCl pH 7.5, 150 mM NaCl). The eluted protein was collected and dialyzed overnight in enzyme buffer (200 mM NaCl, 50 mM Tris-HCl pH 7.5, 1 mM $MgCl_2$, 10% glycerol, 1 mM DTT). Dialyzed enzyme was then stored at 4°C for the short term (< 2 weeks) and −80°C for the long term. Enzyme purity was estimated via SDS-PAGE followed by Coomassie staining.

**Gluconate kinase assay.** Gluconate kinase activity was measured using a continuous measurement of absorbance at 340 nm in a linked assay with the addition of 1 U/mL 6-phosphogluconate dehydrogenase (6-PGDH, Sigma P4553) as an accessory enzyme. Reaction buffer contained 500 $\mu$M NADP+, 200 mM NaCl, 50 mM Tris-HCl (pH 7.5), 1 mM $MgCl_2$, and 1 mM dithiothreitol (DTT). Assays were performed at 37°C with substrates gluconate and ATP alternately supplied in a dilution series to determine Michaelis-Menten constants while the other was supplied in excess, 10 mM gluconate and 10 mM ATP. All data represent means of experimental duplicates of technical replicates. Nonlinear regression was performed using GraphPad Prism to determine apparent Michaelis-Menten constants.

**Virulence studies.** *C. neoformans* H99 and C23 were cultured in YPD for 48 h at 30°C. Harvested cells were washed three times with sterile phosphate-buffered saline (PBS), enumerated with a hemocytometer. Cells were diluted to $1 \times 10^6$ CFU/mL (pulmonary route) and $5 \times 10^4$ CFU/mL (tail vein route) in sterile PBS. A/J female mice (JAX) 8–10 weeks old were inoculated with $5 \times 10^4$ CFU (50 $\mu$L) via intranasal route or $1 \times 10^4$ CFU (200 $\mu$L) via tail vein. Mice were monitored for symptoms; endpoint criteria included head tilt/spinning upon handling, weight loss greater than 10% of baseline, inability to walk or move well enough to eat or drink, and respiratory distress. Survival curves were plotted on a Kaplan-Meier curve, and a log rank test was used to assess the statistical difference of the curves (GraphPad Prism). Fungal burden was assessed on days postinoculation as indicated. Brains were harvested and homogenized in sterile PBS, and 10-fold dilutions were plated on YPD.

**Sporulation and mating.** Crosses were performed on V8 juice agar medium (pH 7.0) in the dark at room temperature (23). Each strain was cocultured with their respective parental strain of the opposite mating locus. Cultures were incubated for 2 weeks and then checked daily up to 8 weeks for spore formation by light microscopy.

**Data analysis and statistics.** All inferential statistics were calculated using GraphPad Prism. Enzyme data were analyzed using a nonlinear regression model for Michaelis-Menten kinetics. Experimental replicates were analyzed with error reported as 95% confidence intervals (CI). Virulence studies were analyzed using the Log-rank Mantel-Cox test. Phylogenetic analysis was performed using Clustal Omega multiple sequence aligner, and phylogenetic tree was constructed using iTOL.

## SUPPLEMENTAL MATERIAL

Supplemental material is available online only.
**SUPPLEMENTAL FILE 1**, PDF file, 0.3 MB.

## ACKNOWLEDGMENTS

We thank Xiaorong Lin and Benjamin Chadwick (Georgia) for helpful discussion. This work was supported by NIH grants 5R01AI147541 (D.J.K.) and T32AI007511 (A.J.J.). The funders had no role in study design, data collection, and interpretation, or the decision to submit the work for publication.

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
