## [Reviewer comments · Microbiology Spectrum]

Microbiology Spectrum

Gluconate kinase is required for gluconate assimilation and sporulation in *Cryptococcus neoformans*.

Andrew Jezewski, Sarah Beattie, Katy Alden, and Damian Krysan

Corresponding Author(s): Damian Krysan, University of Iowa Hospitals and Clinics

Review Timeline:

Submission Date:	January 24, 2022
Editorial Decision:	February 10, 2022
Revision Received:	March 10, 2022
Accepted:	March 19, 2022

Editor: Slavena Vylkova

Reviewer(s): Disclosure of reviewer identity is with reference to reviewer comments included in decision letter(s). The following individuals involved in review of your submission have agreed to reveal their identity: Tong-Bao Liu (Reviewer #3)

Transaction Report:

DOI: <https://doi.org/10.1128/spectrum.00301-22>

February 10, 2022

Dr. Damian J Krysan
University of Iowa Hospitals and Clinics
Division of Pediatrics & Infectious Disease
200 Hawkins Drive, BT 1120
Iowa City, IA 52242

Re: Spectrum00301-22 (Gluconate kinase is required for gluconate assimilation and sporulation in *Cryptococcus neoformans*.)

Dear Dr. Damian J Krysan:

Thank you for submitting your manuscript to Microbiology Spectrum. The reviewers raised some important points that need to be addressed. When submitting the revised version of your paper, please provide (1) point-by-point responses to the issues raised by the reviewers as file type "Response to Reviewers," not in your cover letter, and (2) a PDF file that indicates the changes from the original submission (by highlighting or underlining the changes) as file type "Marked Up Manuscript - For Review Only". Please use this link to submit your revised manuscript - we strongly recommend that you submit your paper within the next 60 days or reach out to me. Detailed instructions on submitting your revised paper are below.

Link Not Available

Sincerely,

Slavena Vylkova

Journals Department
Reviewer comments:

Reviewer #1 (Comments for the Author):

Jezevski et al. provide a genetic and biochemical characterization of IRK3, which had previously been identified in a screen of gene deletion mutants that demonstrated an overcome carbon dioxide stress. Deletion of IRK3 led to a failure to establish infection in the lungs of mice. Biochemical characterization of the recombinantly produced and purified IRK3 enzyme established that it functioned as a 6-phosphogluconate kinase (GNK1), an enzyme in an alternative oxidative phase of the pentose phosphate pathway. In this study, independent deletion mutants of GNK1 were generated and displayed decreased growth on gluconate, but surprisingly did not show altered growth in the presence of carbon dioxide. These deletions mutants were also tested for infectivity using both pulmonary and dissemination mouse models. The authors observed a mixed phenotype with the

deletion mutants that did not correspond to either decreased gluconate utilization or carbon dioxide tolerance. Surprisingly, a GNK1 deletion mutant from a clinical isolate displayed an intermediate virulence phenotype. Lastly, GNK1 was required for sporulation as crosses of the deletion mutant isolates with the parent strain failed to produce spores.

Specific Comments:

(1) In supplemental figure 1, how was the phylogeny constructed? What was the program (statistical methods) used? What are the distances in the phylogeny shown and how many bootstrap replicates?

(2) For the kinetic parameters in Figure 2, the K_m values for ATP and gluconate should be apparent K_m values as Michaelis-Menten kinetics are based on a one substrate-one product model. Did the authors calculate k_{cat} values? Were any other substrates tested?

(3) Line 209: There us a 'we' missing in the sentence.

(4) Because of the difference between the strains used in this study and those in the carbon tolerance genetic screen, it would very difficult (if not impossible) to identify possible second site mutations that could be influencing the phenotypes of the GNK1 deletion strains created in this study. However, would it be worth sequencing the genome of the deletion strains created in this study to identify the differences observed in the mouse studies?

Reviewer #2 (Comments for the Author):

In this manuscript, the authors genetically and biochemically characterized GNK1 (also known as IRK3) as a gluconate kinase, which is required for gluconate assimilation in the human fungal pathogen *Cryptococcus neoformans*. Besides, they demonstrated the key role for Gnk1 in sporulation, thus revealing an importance of gluconate metabolism in formation of meiospores, presumed infectious particles of *C. neoformans*. Furthermore, the authors found that multiple GNK1-deficient mutants generated by them or Yong-Sun Bahn's group had variable virulence and carbon dioxide tolerance. Based on this observation, they propose that second site mutations frequently interact with GNK1 deletion mutations. Overall, the findings are interesting, particularly the characterization of the molecular function of Gnk1 as a gluconate kinase and the previously unrecognized role of gluconate metabolism in sporulation. The authors should address the following comments and questions to improve the manuscript:

1. The gluconate kinase activity of Gnk1 was determined based on in vitro enzyme activity. However, there is no evidence that the phosphorylation of gluconate by Gnk1 is specific. In this regard, other sugars (e.g. glucose) should be tested.
2. There is a relatively strong impact on mating in 'unilateral' crosses (i.e. wild type x mutant), but this may be clearer in bilateral matings (mutant x mutant).
3. Line28 "These data suggest that gluconate metabolism and/or the AOXPPP are critical for sporulation of *C. neoformans*". It would be helpful to test the effect of other mutants of this pathway on sporulation to distinguish whether gluconate metabolism or Gnk1 itself is required for sporulation.
4. The pvalue in Fig. 5A and 5B is not consistent with the description in the main text (line 179, 181).
5. Line 181 "surviving mice infected with the *gnk1* Δ mutant showed significant fungal burden in the brain, indicating that the mutants had disseminated to the brain but did not cause terminal disease". What is the explanation for this phenotype?
6. Some references have an incomplete number of authors, such as ref. 8, 15 and 19.
7. The description of statistical analysis used in this study should be provided in the methods section.

Reviewer #3 (Comments for the Author):

The manuscript authored by Jezewski et al. identified GNK1, a gluconate kinase of the alternative oxidative phase of the pentose phosphate pathway, and characterized its roles in the human fungal pathogen *C. neoformans*. The authors first showed that *C. neoformans* can utilize gluconate as an alternative carbon source in a GNK1-dependent manner. Then the authors' further research showed that the *gnk1* Δ mutants have variable virulence and carbon dioxide tolerance across multiple strains. The author finally showed that the GNK1-deficient strains are unable to sporulate. Overall, it is interesting to find that GNK1 plays a role in both metabolism and copulation/sporulation. The general research strategies and methodologies are basically sound. The study seems to be well executed. However, the current version of the manuscript will not be suitable for publication

in MS because of the following major and minor problems that I would like to address.

1. P162-163, validation of the GNK1 gene knockout mutants. The obtained mutants in this study were verified only by PCR. Sometimes diagnostic PCR does not exclude ectopic insertion during gene knockout. Therefore, the authors had better verify the gene deletion by Southern blotting, especially in the case of different isolates with different phenotypes.
2. How many *gnk1Δ* mutants did the authors obtain by gene knockout? In the growth assay in Fig 4B, the authors used #1-3. However, in Fig 5D of the virulence assay, the authors used #4-5. Why do the authors use different isolates of *gnk1Δ* mutants in different tests?
3. If the mutant showed a distinct phenotype compared to the wild type, it is required to show its complementary strain together to demonstrate that the distinct phenotype is originated from its deletion effect. I recommend that the authors have to include strains exhibiting distinct phenotype and their corresponding complemented strain in the main text and figures.
4. Mating assay. To demonstrate the role of GNK1 in the sexual reproduction of *Cryptococcus*, the authors had better take images with different magnifications and show the details of mating hyphae, basidium and/or basidiospores.
5. Media for mating assay. In the present study, the authors used a V8 medium to induce the mating of *Cryptococcus* strains, and the results showed in Fig 6 was eight weeks post plating on V8 agar. Why did the authors use V8 with a pH of 7.0? Why didn't the authors try MS media? If the MS medium or V8 with different pH (e.g., 5.8) were used, the results might be different!
6. To test the role of GNK1 in sexual reproduction, I suggest that the authors obtain the *gnk1Δ* mutants by knocking out the GNK1 gene in the H99a strain background and then cross with the *gnk1Δ* mutants in the H99alpha background. The complemented strains should be included.
7. Please carefully read through the manuscript to check grammar and spelling errors. There are many grammar errors and several typos, including but not limited to the following:
 - L19: "propose" should be "proposed"
 - L29: add "the" preceding "sporulation"
 - L34: "*C. neoformans*" should be italic, species names should be italic in both the main text, figure legends, and references.
 - L49: add "," after "tolerance"
 - L52: "in vitro" should be italic, similarly hereinafter
 - L63: remove "an" preceding "infection"
 - L65,78: add "the" preceding "carbon"
 - L81: change "which" to "that"
 - L88: remove the repetitive "reducing"
 - L89: remove the repetitive "other"
 - L97: "in vivo" should be italic,
 - L109: remove "that"
 - L114: "At minimum" should be "At a minimum,"
 - L126: change "may" to "might"
 - L138: "6x-His tagged" should be "6x-His-tagged"
 - L150: add "a" preceding "relatively"
 - L200: add "the" preceding "deletion"
 - L217: add "the" preceding "PPP"
 - L218: "increase" should be "increased"
 - L219: "UV exposure" should be "UV-exposure"
 - L224: add "," after "present"
 - L259: "library derived" should be "library-derived"
 - L283: add "was" preceding "also"
 - L365: change "were" to "was"
 - L373: "phosphate buffered" should be "phosphate-buffered"
 - L387: "check daily" should be "checked daily for"

Staff Comments:

Preparing Revision Guidelines

- Point-by-point responses to the issues raised by the reviewers in a file named "Response to Reviewers," NOT IN YOUR COVER LETTER.
- Upload a compare copy of the manuscript (without figures) as a "Marked-Up Manuscript" file.
- Each figure must be uploaded as a separate file, and any multipanel figures must be assembled into one file.
- Manuscript: A .DOC version of the revised manuscript

- Figures: Editable, high-resolution, individual figure files are required at revision, TIFF or EPS files are preferred

Please return the manuscript within 60 days; if you cannot complete the modification within this time period, please contact me. If you do not wish to modify the manuscript and prefer to submit it to another journal, please notify me of your decision immediately so that the manuscript may be formally withdrawn from consideration by Microbiology Spectrum.

Re: Spectrum00301-22 (Gluconate kinase is required for gluconate assimilation and sporulation in *Cryptococcus neoformans*.)

Dear Dr. Damian J Krysan:

Thank you for submitting your manuscript to Microbiology Spectrum. The reviewers raised some important points that need to be addressed. When submitting the revised version of your paper, please provide (1) point-by-point responses to the issues raised by the reviewers as file type "Response to Reviewers," not in your cover letter, and (2) a PDF file that indicates the changes from the original submission (by highlighting or underlining the changes) as file type "Marked Up Manuscript - For Review Only". Please use this link to submit your revised manuscript - we strongly recommend that you submit your paper within the next 60 days or reach out to me. Detailed instructions on submitting your revised paper are below.

Link Not Available

The ASM Journals program strives for constant improvement in our submission and publication process. Please tell us how we can improve your experience by taking this quick Author Survey.

Sincerely,

Slavena Vylkova

Journals Department
Reviewer comments:

Reviewer #1 (Comments for the Author):

Jezewski et al. provide a genetic and biochemical characterization of IRK3, which had previously been identified in a screen of gene deletion mutants that demonstrated an overcome carbon dioxide stress. Deletion of IRK3 led to a failure to establish infection in the lungs of mice. Biochemical characterization of the recombinantly produced and purified IRK3 enzyme established that it functioned as a 6-phosphogluconate kinase (GNK1), an enzyme in an alternative oxidative phase of the pentose phosphate pathway. In this study, independent deletion mutants of GNK1 were generated and displayed decreased growth on gluconate, but surprisingly did not show altered growth in the presence of carbon dioxide. These deletions mutants were also

tested for infectivity using both pulmonary and dissemination mouse models. The authors observed a mixed phenotype with the deletion mutants that did not correspond to either decreased gluconate utilization or carbon dioxide tolerance. Surprisingly, a GNK1 deletion mutant from a clinical isolate displayed an intermediate virulence phenotype. Lastly, GNK1 was required for sporulation as crosses of the deletion mutant isolates with the parent strain failed to produce spores.

Specific Comments:

(1) In supplemental figure 1, how was the phylogeny constructed? What was the program (statistical methods) used? What are the distances in the phylogeny shown and how many bootstrap replicates?

We have added a data analysis and statistical methods section to the text that describes our approach. In short our phylogenetic tree is an alignment based tree that was constructed using Clustal Omega and drawn as a neighbor-joining tree not adjusted for branch lengths using iTOL. We have included branch lengths to the proximal node on the figure.

(2) For the kinetic parameters in Figure 2, the K_m values for ATP and gluconate should be apparent K_m values as Michaelis-Menten kinetics are based on a one substrate-one product model. Did the authors calculate k_{cat} values? Were any other substrates tested?

We have added language to clarify that under classic Michaelis-Menten kinetics we are reporting apparent K_m values. We also clarified that we performed a preliminary assessment of the apparent K_m for each substrate and then followed up with apparent K_m determination in conditions where excess of the alternative substrate was supplied. Under these conditions the apparent K_m are a near perfect approximation of the actual K_m .

We also appreciate the reviewer and other reviewers noting that Gnk1 may accommodate alternative substrates. As with many metabolic enzymes, we expect that Gnk1 would have some ability to accommodate other similar sugars. We provide additional data in Supplemental Figure 3 that supports this. In our Gnk1 activity detection assay we rely on the production of 6P-gluconate and the accessory enzyme 6PGDH. While we cannot directly detect activity from alternative sugar substrates, we can determine if those alternative substrates are competitive with gluconate. We supplied excess alternative sugars (10mM) in conditions where gluconate was supplied at 1/6th the concentration of the K_m for gluconate. This ratio of substrates greatly biases the competing sugar at 20-fold higher than gluconate. Despite very high levels of these competing sugars, we only observed marginal reduction in Gnk1 activity, confirming that Gnk1 has a strong preference for gluconate.

(3) Line 209: There us a 'we' missing in the sentence.

This has been corrected

(4) Because of the difference between the strains used in this study and those in the carbon tolerance genetic screen, it would very difficult (if not impossible) to identify possible second site mutations that could be influencing the phenotypes of the GNK1 deletion strains created in this study. However, would it be worth sequencing the genome of the deletion strains created in this study to identify the differences observed in the mouse studies?

Given that we see a diverse phenotype set across individual Gnk1 deletion strains we anticipate multiple possible genetic interactions, we also anticipate genomic rearrangements and instability may be at play given the sporulation defect. Combined, these concerns reduce the possibility of successful whole genome sequencing. Additionally, it would be unclear if second site mutations are specifically tied to Gnk1 function or an unrelated set of mutations that arose by chance.

Reviewer #2 (Comments for the Author):

In this manuscript, the authors genetically and biochemically characterized GNK1 (also known as IRK3) as a gluconate kinase, which is required for gluconate assimilation in the human fungal pathogen *Cryptococcus neoformans*. Besides, they demonstrated the key role for Gnk1 in sporulation, thus revealing an importance of gluconate metabolism in formation of meiospores, presumed infectious particles of *C. neoformans*. Furthermore, the authors found that multiple GNK1-deficient mutants generated by them or Yong-Sun Bahn's group had variable virulence and carbon dioxide tolerance. Based on this observation, they propose that second site mutations frequently interact with GNK1 deletion mutations. Overall, the findings are interesting, particularly the characterization of the molecular function of Gnk1 as a gluconate kinase and the previously unrecognized role of gluconate metabolism in sporulation. The authors should address the following comments and questions to improve the manuscript:

1. The gluconate kinase activity of Gnk1 was determined based on in vitro enzyme activity. However, there is no evidence that the phosphorylation of gluconate by Gnk1 is specific. In this regard, other sugars (e.g. glucose) should be tested.

We appreciate the reviewer noting that Gnk1 may accommodate alternative substrates. As with many metabolic enzymes, we expect that Gnk1 would have some ability to accommodate other similar sugars. We provide additional data in Supplemental Figure 3 that supports this. In our Gnk1 activity detection assay we rely on the production of 6P-gluconate and the accessory enzyme 6PGDH. While we cannot directly detect activity from alternative sugar substrates, we can determine if those alternative substrates are competitive with gluconate. We supplied excess alternative sugars (10mM) in conditions where gluconate was supplied at 1/6th the concentration of the Km for Gluconate. This ratio of substrates greatly biases the competing sugar at 20-fold higher than gluconate. Despite very high levels of these competing sugars we only observed marginal reduction in Gnk1 activity, confirming that Gnk1 has a strong preference for gluconate.

Additionally, we present evidence that Gnk1 serves a primary role as a gluconate kinase through its highly conserved similarity to other characterized Gnk's and more convincingly show that Gnk1 is required for utilization of gluconate as a carbon source. As with many metabolic enzymes, we did expect that Gnk1 would have some ability to accommodate other similar sugars. Combined this previously provided data and the new data in Supplemental 3 confirm that Gnk1 has a primary biological role in phosphorylation of gluconate. As many of these other sugars have machinery already specific for their phosphorylation, Gnk1 would not be required for that metabolic function in the same way we show Gnk1 is for gluconate phosphorylation.

2. There is a relatively strong impact on mating in 'unilateral' crosses (i.e. wild type x mutant), but this may be clearer in bilateral matings (mutant x mutant).

We have modified figure 6B and the text to more clearly emphasize that we do not observe a defect in mating, as hyphae and basidia are readily formed. The defect remains in sporulation specifically. We do not expect a bilateral (mutant x mutant) mating would result in a defective mating phenotype in addition to a sporulation defect.

3. Line28 "These data suggest that gluconate metabolism and/or the AOXPPP are critical for sporulation of *C. neoformans*". It would be helpful to test the effect of other mutants of this pathway on sporulation to distinguish whether gluconate metabolism or Gnk1 itself is required for sporulation.

We have softened the language in the text to not draw conclusions regarding the role of the AOXPPP in general regarding sporulation, but rather specifically the role of Gnk1.

4. The pvalue in Fig. 5A and 5B is not consistent with the description in the main text (line 179, 181).

This has been corrected.

5. Line 181 "surviving mice infected with the gnk1 Δ mutant showed significant fungal burden in the brain, indicating that the mutants had disseminated to the brain but did not cause terminal disease". What is the explanation for this phenotype?

This is certainly an interesting phenotype. Unfortunately, we are not in a position to be able to clearly answer the mechanism that produces this phenotype. It is possible the mechanism is linked to one of the genetic interactions driving the variability we observe across phenotypes. It should also be noted that a number of mutants have been shown to have significant fungal burden but not induce disease.

6. Some references have an incomplete number of authors, such as ref. 8, 15 and 19.

These have been corrected.

7. The description of statistical analysis used in this study should be provided in the methods section.

Statistical approaches were added to the methods section accordingly.

Reviewer #3 (Comments for the Author):

The manuscript authored by Jezewski et al. identified GNK1, a gluconate kinase of the alternative oxidative phase of the pentose phosphate pathway, and characterized its roles in the human fungal pathogen *C. neoformans*. The authors first showed that *C. neoformans* can utilize gluconate as an alternative carbon source in a GNK1-dependent manner. Then the authors' further research showed that the *gnk1Δ* mutants have variable virulence and carbon dioxide tolerance across multiple strains. The author finally showed that the GNK1-deficient strains are unable to sporulate. Overall, it is interesting to find that GNK1 plays a role in both metabolism and copulation/sporulation. The general research strategies and methodologies are basically sound. The study seems to be well executed. However, the current version of the manuscript will not be suitable for publication in MS because of the following major and minor problems that I would like to address.

1. P162-163, validation of the GNK1 gene knockout mutants. The obtained mutants in this study were verified only by PCR. Sometimes diagnostic PCR does not exclude ectopic insertion during gene knockout. Therefore, the authors had better verify the gene deletion by Southern blotting, especially in the case of different isolates with different phenotypes.

All knockout strains were verified via PCR and several were chosen for Southern blot results in figure 5E, including strains with different phenotypes.

2. How many *gnk1Δ* mutants did the authors obtain by gene knockout? In the growth assay in Fig 4B, the authors used #1-3. However, in Fig 5D of the virulence assay, the authors used #4-5. Why do the authors use different isolates of *gnk1Δ* mutants in different tests?

We obtained many and confirmed nearly a dozen mutants. All confirmed mutants were tested for CO₂ tolerance and a representative plate was shown. All our generated and tested mutants exhibited CO₂ tolerance. Only two mutants in the H99 background and one in the C23 background, in consideration of animal stewardship, were chosen at random to carry out animal experiments. This random selection revealed different phenotypes across these strains.

3. If the mutant showed a distinct phenotype compared to the wild type, it is required to show its complementary strain together to demonstrate that the distinct phenotype is originated from its deletion effect. I recommend that the authors have to include strains exhibiting distinct phenotype and their corresponding complemented strain in the main text and figures.

We extensively discuss why genetic complementation is not appropriate in this case. In short, we are clear that we expect the phenotypes to be the result of genetic interactors. These interactors are likely different across isolates and will provide different outcomes resulting from complementation experiments depending on the necessity vs sufficiency of *gnk1* mutants in the respective background. Ultimately, these experiments will not identify the interacting loci.

4. Mating assay. To demonstrate the role of GNK1 in the sexual reproduction of *Cryptococcus*, the authors had better take images with different magnifications and show the details of mating hyphae, basidium and/or basidiospores.

The defect is in sporulation rather than sexual reproduction. We show the relevant images that others have shown for similar phenotypes and have cited them accordingly.

Feretzi, Marianna, and Joseph Heitman. 2013. Genetic circuits that govern bisexual and unisexual reproduction in *Cryptococcus neoformans*. *PLoS genetics* 9.8: e1003688.

Fan C-L, Han L-T, Jiang S-T, Chang A-N, Zhou Z-Y, Liu T-B. 2019. The Cys2His2 zinc finger protein Zfp1 regulates sexual reproduction and virulence in *Cryptococcus neoformans*. *Fungal Genet Biol* 124:59–72.

Liu L, He G-J, Chen L, Zheng J, Chen Y, Shen L, Tian X, Li E, Yang E, Liao G. 2018. Genetic basis for coordination of meiosis and sexual structure maturation in *Cryptococcus neoformans*. *Elife* 7:e38683.

Bahn Y-S, Cox GM, Perfect JR, Heitman J. 2005. Carbonic anhydrase and CO₂ sensing during *Cryptococcus neoformans* growth, differentiation, and virulence. *Curr Biol* 15:2013–2020.

5. Media for mating assay. In the present study, the authors used a V8 medium to induce the mating of *Cryptococcus* strains, and the results showed in Fig 6 was eight weeks post plating on V8 agar. Why did the authors use V8 with a pH of 7.0? Why didn't the authors try MS media? If the MS medium or V8 with different pH (e.g., 5.8) were used, the results might be different!

We used the standard V8 media as has been previously used to identify mutants with sporulation defects. We added the appropriate citation to our methods. Alternative media is usually considered when mating is unable to occur. Our mutants are able to mate and form hyphae, but not able to sporulate. Therefore, V8 media sufficiently induces the conditions necessary for sexual reproduction and previous sporulation defects have been assessed in the media type.

Feretzi, Marianna, and Joseph Heitman. 2013. Genetic circuits that govern bisexual and unisexual reproduction in *Cryptococcus neoformans*. *PLoS genetics* 9.8: e1003688.

6. To test the role of GNK1 in sexual reproduction, I suggest that the authors obtain the *gnk1Δ* mutants by knocking out the GNK1 gene in the H99a strain background and then cross with the *gnk1Δ* mutants in the H99alpha background. The complemented strains should be included.

We report that *gnk1Δ* mutants are able to form hyphae as represented in Fig. 6B. Therefore, mating has already occurred (hyphae are 2N) but a sporulation defect remains. A *gnk1Δ* mutant in the opposite mating type would result in the same (2N) state and thus would not change the inability to sporulate.

7. Please carefully read through the manuscript to check grammar and spelling errors. There are many grammar errors and several typos, including but not limited to the following:

L19: "propose" should be "proposed"

L29: add "the" preceding "sporulation"

L34: "*C. neoformans*" should be italic, species names should be italic in both the main text, figure legends, and references.

L49: add ", " after "tolerance"

L52: "in vitro" should be italic, similarly hereinafter

L63: remove "an" preceding "infection"

L65,78: add "the" preceding "carbon"

L81: change "which" to "that"
L88: remove the repetitive "reducing"
L89: remove the repetitive "other"
L97: "in vivo" should be italic,
L109: remove "that"
L114: "At minimum" should be ""At a minimum,"
L126: change "may" to "might"
L138: "6x-His tagged" should be "6x-His-tagged"
L150: add "a" preceding "relatively"
L200: add "the" preceding "deletion"
L217: add "the" preceding "PPP"
L218: "increase" should be "increased"
L219: "UV exposure" should be "UV-exposure"
L224: add "," after "present"
L259: "library derived" should be "library-derived"
L283: add "was" preceding "also"
L365: change "were" to "was"
L373: "phosphate buffered" should be "phosphate-buffered"
L387: "check daily" should be "checked daily for"

We greatly appreciate the reviewer's thorough reading of the manuscript and identifying the above errors, we have corrected them accordingly.

March 19, 2022

Dr. Damian J Krysan
University of Iowa Hospitals and Clinics
Division of Pediatrics & Infectious Disease
200 Hawkins Drive, BT 1120
Iowa City, IA 52242

Re: Spectrum00301-22R1 (Gluconate kinase is required for gluconate assimilation and sporulation in *Cryptococcus neoformans*.)

Dear Damian,

I am happy to inform you that your manuscript has been accepted, and I am forwarding it to the ASM Journals Department for publication. You will be notified when your proofs are ready to be viewed.

Sincerely,

Slavena Vylkova
Editor, Microbiology Spectrum
